# Ionic Liquid/Deep Eutectic Solvent-Mediated Ni-Based Catalysts and Their Application in Water Splitting Electrocatalysis

**Chenyun Zhang [1], Te Bai [1], Yefan Sun [1], Bingwei Xin [2,*] and Shengnan Zhang [2]**

1   School of Intelligent Manufacturing, Wuxi Vocational College of Science and Technology, Wuxi 214028, China
2   College of Chemistry and Chemical Engineering, Dezhou University, No. 566 West University Road Decheng District, Dezhou 253023, China
*   Correspondence: xinbingwei@dzu.edu.cn; Tel.: +86-136-8534-5517

**Abstract:** Nickel-based electrocatalysts have been widely used to catalyze electrocatalytic water splitting. In order to obtain high-performance nickel-based electrocatalysts, using ionic liquids and deep eutectic solvents mediated their preparation has received increasing attention. Firstly, ionic liquids and deep eutectic solvents can act as media and templates for the preparation of Ni-based nanomaterials with novel structures and excellent catalytic activity. Secondly, ionic liquids and deep eutectic solvents can be employed as reactants to participate the synthesis of catalysts. Their participation not only increase the catalytic performance, but also simplify the reaction system, improve reproducibility, reduce emissions, and achieve atomic economy. On the basis of the work of our group, this review gives a detailed description of the impressive progress made concerning ionic liquids and deep eutectic solvents in the preparation of nickel-based electrocatalysts according to their roles. We also point out the challenges and opportunities in the field.

**Keywords:** nickel-based electrocatalyst; ionic liquid; deep eutectic solvent; water splitting electrocatalysis

## 1. Introduction

The application of traditional energy leads to a large amount of carbon emissions and energy depletion. Therefore, traditional fossil fuels such as coal, oil and natural gas will be replaced by green energy. As one of the important clean energy sources in the future, hydrogen has the advantages of highly availability, recyclability and so on. Therefore, the preparation of hydrogen has become an important research issue in the energy field [1–4]. Electrochemical water splitting is one of the most promising ways for $H_2$ production in industry. Water splitting is divided into two half reactions, hydrogen evolution reaction (HER) and oxygen evolution reaction (OER) [5,6]. Regrettably, the slow kinetics of water splitting seriously hinders its industrialization. Therefore, electrocatalysts play a key role in energy conversion technology. Generally, precious metals-based catalysts, such as Pt, $RuO_2$, and $IrO_2$, have always been chosen as catalysts for water splitting in industrial production. However, their low reserves and high prices hinder their large-scale application. Therefore, non-precious transition metal-based catalysts received considerable attention, such as Fe, Co, Ni, Mn, Mo, etc. In this review, we focus on nickel-based catalysts. Nickel element has d orbital electrons. It can provide empty orbitals to act as electrophiles in chemical reactions to form intermediate products. Therefore, nickel-based catalysts can reduce the activation energy of reactions, promoting the reactions. They are one kind of the most active catalysts for water electrolysis [7–10]. A large number of nickel oxides, hydroxides, sulfides, phosphides, etc. have been synthesized by various strategies. However, it is found that there are some limitations in traditional synthesis methods. Firstly, to obtain novel morphologies and excellent performance of Ni-based catalysts, soft or hard templates (such as surfactants) are generally needed in the preparing process. However, templates will result in by-products and poor reproducibility while cumbersome post-processing. Secondly, the preparation of Ni-based catalysts is often carried out at high temperature

with high pressure, making reaction process dangerous and harmful. Thirdly, traditional sulfur and phosphorus sources for the synthesis of $NiP_x$ or $NiS_x$ are generally flammable, explosive and toxic. Under such circumstances, ionic liquids (ILs) and deep eutectic solvents (DESs) are introduced into the synthesis of nickel-based electrocatalysts.

Ionic liquids, as the name suggests, are salts that exhibit liquid phase below 100 °C. They are composed entirely of ions with relatively large volume differences, making them difficult to densely pack or crystallize [11–13]. Special compositions and structural features lead ILs to possess unique physical and chemical properties, such as low melting point and nonvolatility, strong conductivity and so on. As diverse functional materials, ILs have made remarkable achievements in the field related to chemistry [14–16]). Especially in the field of nanomaterial preparation, they provide growth environment that is different from conventional solvents, changing the growth processes of catalysts. Therefore, ILs have the ability to shape the morphology of nanomaterials. In addition, Ils can stable inorganic micro/nanostructures, which are recognized as good nanoparticle stabilizers [17,18]. The requirement for novel micro/nanomaterials has stimulated the drastic development of Ils in the preparation of catalysts for electrocatalytic water splitting. However, with the in-depth and extensive research of Ils, there are several problems that trouble people, including the high price, difficult purification and gradually discovered toxicity [19]. Therefore, deep eutectic solvents as alternatives to Ils have attracted attention.

Deep eutectic solvents, just like Ils, are the other kind of materials with relatively low freezing point. The concept of DESs was proposed by Abbott and co-workers in 2003 [5,20]. DESs are two-component or three-component deep eutectic mixtures formed by hydrogen bond acceptors (HBAs) and hydrogen bond donors (HBDs) in a certain molar ratio through hydrogen bonds. It is these hydrogen bond interactions that induce the freezing point of DESs to be much lower than that of any component, driving the mixtures to liquid. HBAs usually include quaternary ammonium salts, quaternary phosphonium salts, metal halides, etc., while HBDs generally refer to amides, carboxylic acids, polyols and so on. Currently, choline chloride ($HOCH_2CH_2N\,(CH_3)_3{}^+Cl^-$, ChCl)-based DESs are the most used type of DESs, such as ChCl/urea, ChCl/ethylene glycol (EG), ChCl/glycerol, and so on. DESs have similar properties to ILs, which possess low freezing point, great solubility, good electrical conductivity, excellent thermal stability and high viscosity, etc. However, DESs have more advantages than ILs, such as facile purification, low price, good biodegradability and biocompatibility. So DESs are more promising in industry than ILs. Since DESs were first discovered, they have been quickly applied to the fields of organic synthesis, extraction, nanomaterial preparation, catalysis, energy conversion, and other fields [21–24].

In the field of catalysis, research on ILs and DESs in the preparation of catalysts has achieved dramatic development. They are already recognized as excellent reaction media. Owing to their special physical and chemical properties, they can change the reaction mechanism and product structures, and obtain products that can't get in traditional media. In addition, with the increasing awareness of environmental protection, reducing emissions has become an important strategy. For this purpose, people are devoted to the study of simplifying the reactant components through a strategy in which solvent is directly involved in the reaction as a reactant. Fortunately, ILs and DESs have been found to act simultaneously as both solvent and reactant. Therefore, ILs and DESs have the ability to play several roles in preparation of catalysts, including media, templates, reactants and so on [25–31].

Our research group has been focusing on the preparation of IL-/DES-derived Ni-based catalysts and exploring their activity in the field of water electrolysis. In this review, the research progress of using ILs and DESs to prepare nickel-based catalysts for water splitting is overviewed in detail combined with our work. This review is divided into four sections. In Section 1, Introduction section, the background of the preparation of Ni-based catalysts with ILs and DESs is presented. Section 2 describes the achievements of Ni-based catalysts in ILs. In Section 3, we summarize the application of DESs in the preparation of Ni-based catalysts. In Section 4, we offer conclusions about this review, while analyze the issues and

prospects for the next research in this domain. Considering the rapid developments in this topic, the achievements of IL- and DES-mediated Ni-based catalysts presented here are current as of August 2022.

## 2. Using Ionic Liquids to Prepare Nickel-Based Electrocatalysts

The high ionic strength and viscosity as well as abundance of hydrogen bonds in ILs make the microenvironment different from conventional solvents [32]. In the following description, we discuss progress in this area according to different roles of ILs. We summarize the relevant literature in Table 1.

### 2.1. Ionic Liquids as Reaction Solvents and Templates to Prepare Nickel-Based Electrocatalysts

ILs act widely as solvents and soft templates for nanostructures. The advantages are as follows. Firstly, ILs are almost nonvolatile and have highly thermal stability. Therefore, they can maintain normal pressure or even vacuum at high temperature in reactions, which not only reduce the production of toxic gases but also avoid the danger of high-pressure reactions. Secondly, ILs can dissolve materials that are difficult to dissolve in conventional solvents, such as metals, metallic oxides, and so on, so that the reactions can proceed homogeneously [25–27]. Thirdly, ILs have special physicochemical properties and structures. On the one hand, ILs have greater viscosity, higher polarity, lower surface tension than aqueous solution or conventional organic solvents. On the other hand, they can spontaneously form extended hydrogen bonds. Therefore, ILs can regulate the morphology of inorganic nanomaterials, while tune the nucleation and growth of catalysts. Fourthly, the microwave absorption and large electrochemical window of ILs make them suitable for microwave and electrodeposition synthesis. Various nickel-based catalysts have been successfully prepared in ILs.

Nickel-based oxides are commonly catalysts for electrolysis of water. Using ILs as media or solvent auxiliaries is an effective measure to modulate their morphology. In the mixed solvent of water and ethanol, 1-ethyl-3-methylimidazolium nitrate ([EMIM]NO$_3$) could assist the reaction between Fe(NO$_3$)$_3$, Ni(NO$_3$)$_2$ and urea to obtain a mixture of r-Fe$_2$O$_3$, NiO and NiOOH (abbreviated as IL-NiO·Fe$_2$O$_3$) [33]. Meanwhile, the hydrophobicity of [EMIM]$^+$ rooted in alkyl side chains could reduce the surface tension of solvent, regulating the formation of nanoparticles. Therefore, IL-NiO·Fe$_2$O$_3$ particles showed superior structure to these obtained from the system without IL, which had the characterics of high pores, small particle sizes, large specific surface areas and good dispersibility. As a task specific IL, IL containing -NH$_2$ can controllably tune the structure of catalysts. In methanol/water mixed solvent containing Ni(NO$_3$)$_2$·6H$_2$O or the mixture of Fe(NO$_3$)$_3$·9H$_2$O, Co(NO$_3$)$_2$·6H$_2$O, Ni(NO$_3$)$_2$·6H$_2$O (1:1 molar ratio), the addition of 1-propyl-3-methylimidazolium tetrafluoroborate ([PMIM]BF$_4$)-NH$_2$ and NaOH achieved a novel and versatile one-step method to synthesize IL-modified amorphous hydroxide nanoparticles [34]. IL-NH$_2$ led to a large number of vacancies. Meanwhile, the IL layer immobilized on the solid support could provide interfacial microenvironment, which kept relative higher OH$^-$ concentration on the oxide surface particles and further facilitated the extrusion of the resulting O$_2$. These catalysts were beneficial to promote OER activity. IL-NiO·Fe$_2$O$_3$ particles revealed more excellent catalytic activity for OER than NiO·Fe$_2$O$_3$ without the involvement of IL. Amorphous NiFe hydroxide exhibited high OER activity ($\eta_{10}$ (the overpotential corresponding to a current density of 10 mA cm$^{-2}$)) = 300 mV, Tafel slope = 54.4 mV dec$^{-1}$) (Figure 1).

Nickel phosphides are one of kind of the most studied metal phosphides. Their structures and surface area can also be tuned by ILs. Zheng designed a system containing NiCl$_2$·6H$_2$O, red phosphorus, diethanolamine and different amounts of 1-butyl-3-methylimidazolium chloride ([BMIM]Cl) [35]. [BMIM]$^+$ cation could interact with phosphorous anion induced by diethanolamine, which slowed down the diffusion rate of nickel cation and the decomposition rate of red phosphorus, increasing the porosity and pore size of as-prepared catalyst. Meanwhile, the amount of IL had a great influence on the number

of pores. Therefore, the participation of IL could enlarge the surface area, improving the catalytic performance. The $Ni_2P$ obtained in the optimal amount of [BMIM]Cl possessed large specific surface areas with big pore volume, which exhibited excellent OER catalytic performance ($\eta_{10}$ = 246 mV, Tafel slope = 46 mV dec$^{-1}$).

ILs also have a great influence on the crystal phase of catalysts in addition to the morphologies. Studies have revealed that the larger the proportion of phosphorus atoms in the phosphides, the more beneficial to catalyze HER. The reason is that the partial negative charge on the phosphorus atom not only regulate the surface electronic structure of transition metals but also conducive to capture protons and desorb hydrogen [36–39]. However, conventional solvents tend to give mixed phases. For example, when using octadecene (an organic solvent) as a solvent, the mixture of $Ni_2P$ and $Ni_{12}P_5$ was obtained in the system involving $PPh_3$, $Ni(acac)_2$ and oleylamine. $Ni_{12}P_5$, the metal-rich phosphides, would definitely reduce the HER performance of $Ni_2P$. However, [BMIM]Tf$_2$N was used as a medium to induce the preparation of pure $Ni_2P$ nanocrystals under same condition as that in octadecene, as shown in Figure 2 [40]. That is, participation of IL solved the problem of impure crystal phase. This is because different media lead to different growth processes of nickel phosphides. In 1-ethyl-3-methylimidazolium bis (trifluoromethylsulfonyl)imide ([BMIM]Tf$_2$N) system, [BMIM]$^+$ cations were adsorbed on the surface of $Ni_2P$ crystal nuclei, which in turn affected the nucleation and growth of phosphides. Nickel phosphides had no crystal phase transformation during the growth process so that a single crystal phase was obtained. However, non-polar octadecene could not be adsorbed on the surface of nickel phosphides, undergoing a transformation from $Ni_{12}P_5$ to $Ni_2P$ during the phosphating process. Therefore, IL could effectively adjust the crystal phase of nickel phosphides and optimized the catalytic performance.

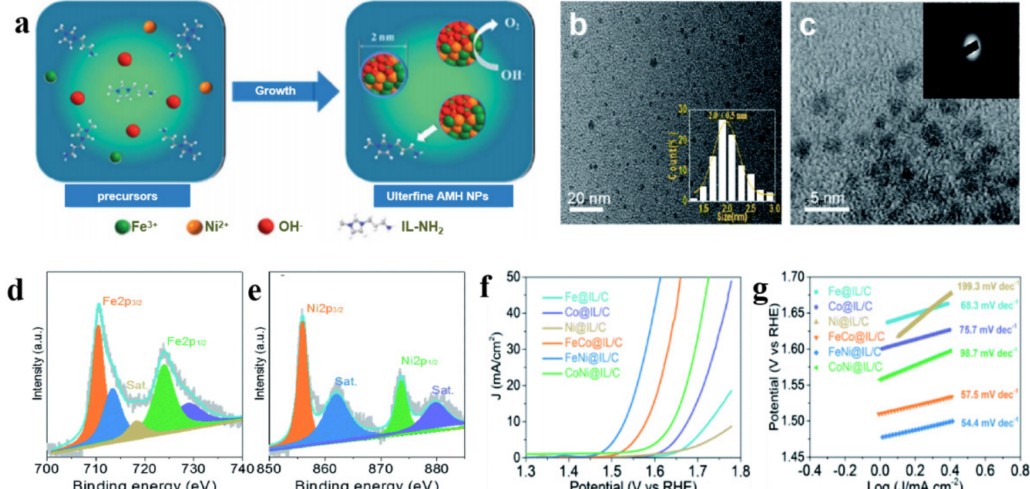

**Figure 1.** Strategy for the synthesis of amorphous NiFe hydroxide (**a**), HRTEM image (**b**), HAADF-STEM image (inset: the corresponding SAED pattern) (**c**) and high-resolution XPS spectra of Fe$_2$p (**d**), Ni$_2$p (**e**) of amorphous NiFe hydroxide, polarization curves (**f**) as well as Tafel slopes (**g**) of amorphous NiFe hydroxide and the comparative electrocatalysts for OER in alkaline solution (Reprinted with permission from Ref. [34], 2020, Royal Society of Chemistry).

**Table 1.** Summary of HER, OER and Water Splitting Performance of Catalysts Involved in the Section 2.

| Catalyst | Morphology | Applied IL | Synthesis Method | Synthesis Condition | Catalytic Performance | | | | | | | | Ref. |
|---|---|---|---|---|---|---|---|---|---|---|---|---|---|
| | | | | | HER | | | OER | | | Water Splitting | | |
| | | | | | Electrolyte | $\eta$ (mV)@Current Density (mA cm$^{-2}$) | Tafel Slope (mV dec$^{-1}$) | Electrolyte | $\eta$(mV)@Current Density (mA cm$^{-2}$) | TafelSlope (mV dec$^{-1}$) | Electrolyte | Potential (V)@Current Density (mA cm$^{-2}$) | |
| $NiO \cdot Fe_2O_3$ | Nanoparticles | [EMIM]NO$_3$ | Hydrothermal method | 180 °C, 12 h | – | – | – | 1 M KOH | – | – | – | – | [33] |
| NiFe hydroxides | Nanoparticles | [PMIM]BF$_4$ | Coprecipitation | Room temperature, 24 h | – | – | – | 1 M KOH | 300@10 | 54.4 | – | – | [34] |
| $Ni_2P$ | Big bulk grains with surface and interior thickly dotted with holes | 1-butyl-3-methylimidazolium chloride ([BMIM]Cl) | Self-templated wet-chemical method | 120 °C, 12 h | – | – | – | 1 M KOH | 246@10 | 46 | – | – | [35] |
| $Ni_{12}P_5$ | Hollow structure | – | Reflow heating | 320 °C, 2 h | 0.5 M H$_2$SO$_4$ | 208@10 | 75 | – | – | – | – | – | [36] |
| $Ni_2P$ | Rough surface morphology | – | Ball milling method | 500 rpm, 6 h | 0.5 M H$_2$SO$_4$ | – | 79 | – | – | – | – | – | [37] |
| $Ni_2P$ | Netlike structure | – | Hydrothermal method | 140 °C, 15 h | 1 M KOH | 85@10 | – | 1 M KOH | 260@10 | 112 | – | – | [38] |
| $Ni_{12}P_5$ | Netlike structure | – | Hydrothermal method | 200 °C, 15 h | 1 M KOH | 170@10 | – | 1 M KOH | 240@10 | 106 | – | – | [38] |
| S-NiFeP | Nanoparticles | – | Hydrothermal and annealing methods | 150 °C, 24 h; 300 °C, 2 h | 1 M KOH | 56@10 | 38 | 1 M KOH | 201@10 | 41.2 | 1 M KOH | 1.5@10 | [39] |
| $Ni_2P$ | Nanoparticles | [BMIM]Tf$_2$N | Reflow heating | 310 °C, 30 min | 0.5 M H$_2$SO$_4$ | 107@10 | 70 | – | – | – | – | – | [40] |
| NiZn | Nanowires | [BMP]Tf$_2$N | Electrodeposition and galvanic displacement reaction | −0.67 V (vs. Ag/Ag$^+$) and 80 °C | – | – | – | 1 M NaOH | – | – | – | – | [41] |
| NiFeP | Nanosheets | – | Annealing method | 300 °C, 2 h | 1 M KOH | 130@20 | 78 | – | – | – | – | – | [42] |
| NaYF$_4$ | Spherical morphology | [BMIM][BF$_4$] | Ionothermal method | 200 °C, 12 h | – | – | – | – | – | – | – | – | [43] |
| P,F-Ni$_{1.5}$Co$_{1.5}$N | Nanorods | [BMIM]PF$_6$ | Hydrothermal method and annealing process | 150 °C, 10 h and 400 °C, 3 h in NH$_3$ atmosphere | – | – | – | 1 M KOH | 280@10 | 66.1 | – | – | [44] |
| Cu$_x$Ni$_{1-x}$ | Flower-like morphology | [BMIM]PF$_6$ | Hydrothermal method | 200 °C, 20 h | 1 M KOH | 88@10 | 91 | 1 M KOH | 198@10 | 76 | 1 M KOH | 1.58@10 | [45] |
| $Ni_2P$ wrapped by carbon fiber | Nanoparticles | [BMIM]PF$_6$ | Inkjet printing technology | Using a commercial electrohydro-dynamic inkjet printer | 1 M KOH | 117@10 | 92.0 | – | – | – | – | – | [46] |
| NiP$_2$ | Nanoparticles | [P$_{4444}$]Cl | Microwave | 50 W, 1 min 50 s | 0.5 M H$_2$SO$_4$ | 102@10 | 46 | – | – | – | – | – | [47] |

**Table 1.** *Cont.*

| Catalyst | Morphology | Applied IL | Synthesis Method | Synthesis Condition | Catalytic Performance | | | | | | | | | Ref. |
|---|---|---|---|---|---|---|---|---|---|---|---|---|---|---|
| | | | | | HER | | | OER | | | | Water Splitting | | |
| | | | | | Electrolyte | $\eta$ (mV)@ Current Density (mA cm$^{-2}$) | Tafel Slope (mV dec$^{-1}$) | Electrolyte | $\eta$(mV)@ Current Density (mA cm$^{-2}$) | TafelSlope (mV dec$^{-1}$) | Electrolyte | Potential (V)@Current Density (mA cm$^{-2}$) | | |
| Co$_2$P | Shuttle shaped morphology | Trihexyl(tetradecyl) phosphonium tetrachlorocobaltate ([P$_{66614}$]$_2$[CoCl$_4$]) | Annealing method | 400 °C, 2 h | 0.5 M H$_2$SO$_4$ | 150@10 | 47 | – | – | – | – | – | – | [48] |
| Co$_2$P | Spherical morphology | tetrabutylphosphonium tetrachlorocobaltate(II) ([P$_{4444}$]$_2$[CoCl$_4$]) | Microwave | 6 min | 0.5 M H$_2$SO$_4$ | 135@10 | 58 | – | – | – | – | – | – | [49] |
| Fe$_2$P | Spherical morphology | trihexyl(tetradecyl) phosphonium tetrachloroferrate ([P(C$_6$H$_{13}$)$_3$C$_{14}$H$_{29}$][FeCl$_4$]) | Annealing method | 400 °C, 2 h | 0.5 M H$_2$SO$_4$ | 115@10 | 68 | – | – | – | – | – | – | [50] |
| Ni$_2$P$_4$O$_{12}$ | Nanoparticles | Octylamine/ hypophosphorous | Annealing method | 400 °C, 2 h | 0.5 M H$_2$SO$_4$ | 116@10 | 97 | – | – | – | – | – | – | [51] |

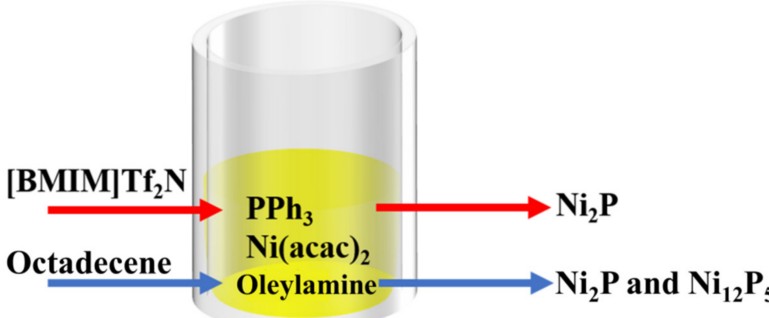

**Figure 2.** Pure $Ni_2P$ and the mixture of $Ni_2P/Ni_{12}P_5$ obtained using [BMIM]$Tf_2N$ or octadecene as medium, respectively [40].

ILs are suitable to electrodeposition processes due to their excellent conductivity and large electrochemical window. Using ILs as electrolyte, their relatively high viscosity and ionic force with metal ions will realize the controllable preparation of catalysts under certain conditions. The fabrication of hollow NiZn nanotubes in ILs is a good example [41]. Entangled porous Zn nanowires were prepared by applying electrodeposition in amide type IL, 1-butyl-3-methylpyrrolidinium bis(trifluoromethyl)sulfonyl)imide ([BMP]$Tf_2N$) containing $ZnCl_2$. Then using the obtained Zn nanowires as sacrificial templates, hollow NiZn nanotubes were gotten through the galvanic displacement reaction (NGR) between $NiCl_2$ and Zn. NGR is a kind of reaction that uses the inherent potential difference between different metals as driving force to prepare metal elements or compounds. Compared with other processes, NGR does not need to consume any energy [42]. The relatively high viscosity and cations of IL were crucial to successfully retain the template structure. Using water or ChCl/EG to replace the IL as comparison experiment could not obtain the above morphology. IL to play a key role in the controllable synthesis of hollow NiZn nanotubes.

### 2.2. Ionic Liquids as Reaction Reagents to Prepare Nickel-Based Electrocatalysts

ILs can decompose at high temperatures although good thermal stability is their one of the features. This performance enables ILs to participate in catalyst formation as reactants [28–31]. Being benefit from the "designability" of ILs, the rational design of anions and cations in ILs will make them good sources of N, C, P, S, or other heteroatoms, directly forming nitrides, carbides, or other heteroatom-doped catalysts [43]. The addition of heteroatoms can change the surface morphologies and electronic structures of catalysts, reduce the onset potential, and improve the catalytic activities.

$PF_6^-$ is a commonly anion in ILs. $PF_6$-based ILs can provide phosphorus atoms under certain conditions. When their amount is relatively small, it will lead to the formation of phosphorus-doped metal catalysts. Zhang and Wang group conducted detailed studies on this aspect [44]. They injected an appropriate amount of 1-butyl-3-methylimidazolium hexafluorophosphate ([BMIM]$PF_6$) into water/ethanol solution containing $Ni^{2+}:Co^{2+}$ with a mass ratio of 1:1 and hydrazine hydrate. P,F co-doped $Ni_{1.5}Co_{1.5}N$ nanorods were synthesized by solvothermal method and annealing in $NH_3$ atmosphere (Figure 3a–d). P and F atoms were offered from [BMIM]$PF_6$. N had two sources, including IL containing N element in cation and $NH_3$. IL-involved synthesis strategy was simple and convenient, and could integrate the synergistic effects of various active components. Thanks to multiple factors involving the surface heteroatom doping, one-dimensional mesoporous structure, the synergistic effect between transition metals, this P,F co-doped $Ni_{1.5}Co_{1.5}N$ showed excellent OER activity. The $\eta_{10}$ was 280 mV, while the Tafel slope was 66.1 mV dec$^{-1}$ (Figure 3e–g). This research group continuously used [BMIM]$PF_6$ to prepare N, P and F doped $Cu_{0.81}Ni_{0.19}$ alloys with 3D flower shape through adding copper ion and Ni foam (NF) into system based on [BMIM]$PF_6$ [45]. IL as a reducing agent and heteroatom dopant changed the morphology of NF from irregular particles to octahedron under hydrothermal conditions. The presence of IL not only induced the flower-like morphological character-

istics of $Cu_{0.81}Ni_{0.19}$ alloys, but also tailored its interfacial electronic structure. The alloy was a dual-function catalyst in alkaline medium, the $\eta_{10}$ values of which were 88 mV and 198 mV for HER and OER, respectively.

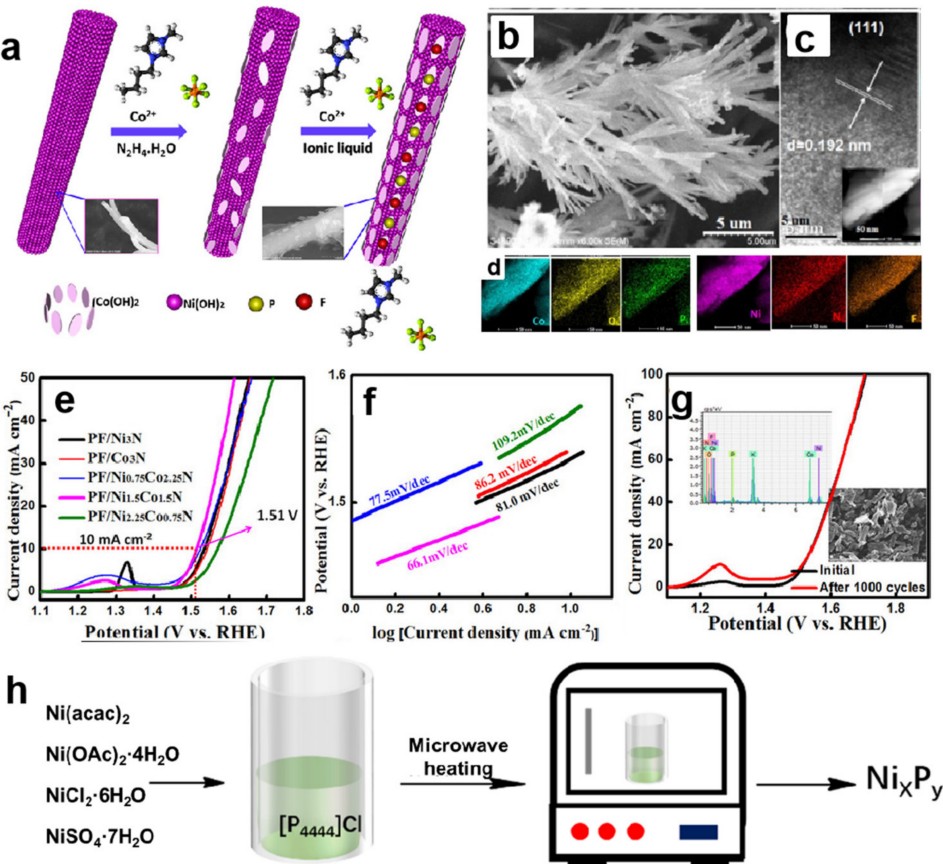

**Figure 3.** Schematic illustration of the fabrication processes of $PF/Ni_{1.5}Co_{1.5}N$ (**a**), SEM image (**b**), HR-TEM image (inset: the corresponding HAADF-STEM) (**c**) and mapping results (**d**) of $PF/Ni_{1.5}Co_{1.5}N$, polarization curves (**e**) as well as Tafel slopes (**f**) of $PF/Ni_{1.5}Co_{1.5}N$ and the comparative catalysts for OER in alkaline solution, LSV curves and morphology of the $PF/Ni_{1.5}Co_{1.5}N$ catalyst before and after CV testing of 1000 cycles (**g**) (Reprinted with permission from Ref. [44], 2017, Wiley); Schematic diagram to prepare nickel phosphide in $[P_{4444}]Cl$ IL (**h**) [47].

When the amount of ILs containing P in anions is large, it can be used as a phosphorus source to prepare metal phosphides. The decomposed phosphorus element from $[BMIM]PF_6$ induced the formation of metal phosphide with N,P co-doped carbon [46]. ILs as P source contributed to expose more active sites and increase electron transfer rate. Meanwhile, $[BMIM]PF_6$ prevented the aggregation of nanoparticles. Therefore, the prepared catalyst had high HER performance with long-time stability in 0.5 M $H_2SO_4$ and 1 M KOH. Among them, $Ni_2P$ with N,P co-doped carbon only required overpotentials of 102 mV to obtain a current density of 10 mA cm$^{-2}$ in acidic environment and 117 mV to obtain the same current densities in alkaline media.

Recently, our research group have explored the synthesis method of nickel phosphides based on quaternary phosphonium salt [47]. We found that $[P_{4444}]Cl$ IL could serve as template and phosphorus source. It reacted with different nickel salts ($Ni(acac)_2$, $Ni(OAc)_2\cdot4H_2O$, $NiCl_2\cdot6H_2O$ and $NiSO_4\cdot7H_2O$), and could complete quickly reaction within 2 min by using microwave heating. The as-prepared small nickel phosphide particles with a single crystal phase were showed in Figure 3h. The anion of nickel salt had a remarkable effect on the composition and morphology of nickel phosphide. When the anion was acac$^-$ or OAc$^-$, $Ni_2P$ nanoparticles could be obtained with an average size of

$12.0 \pm 3.3$ or $30.24 \pm 8.0$ nm, respectively. In contrast, $Ni1_2P_5$ nanoparticles were obtained when the anion was $Cl^-$ or $SO_4^{2-}$, which presented $122.5 \pm 5.0$ nm particles or sponge-like nanocrystals, respectively. All of the synthesized nickel phosphides had excellent electrocatalytic performance for hydrogen production under acidic conditions. Among them, $Ni_2P$ prepared by $Ni(acac)_2$ had an overpotential of 102 mV to achieve 10 mA cm$^{-2}$ and the Tafel slope of only 46 mV dec$^{-1}$. The catalytic activity was not only higher than other nickel phosphides prepared in this experiment but also better than most reported nickel-based hydrogen evolution electrocatalysts. We speculate that ILs as reactive reagents can lead the active species to uniformly disperse in products, inducing the structures with long-range order. Since then, researchers have launched a positive study on the application of phosphorus-containing ILs as phosphorus sources. Li et al. have been successfully prepared cobalt phosphides and iron phosphides by using similar ILs [48–50].

Subsequently, our group explored other new phosphorus-contained ILs to synthesize high-performance Ni-based catalysts. We designed a proton-type IL, octylamine/hypophosphorous acid through mixing octylamine and hypophosphorous acid [51]. Compared with other ILs, the proton-type IL is cheap and easy to obtain, being conducive to large-scale production. Calcining this proton-type IL with $NiCl_2$ together obtained $Ni_2P_4O_{12}$ with high crystallinity. The prepared $Ni_2P_4O_{12}$ had good hydrogen evolution performance. Under alkaline conditions, the overpotential was only 116 mV at a current density of 10 mA cm$^{-2}$, while the Tafel slope was 97 mV dec$^{-1}$.

Obviously, ILs can act as reactants to prepare nickel-based electrocatalysts under appropriate conditions. This synthesis strategy has the following advantages: (1) Compared with traditional heteroatom sources (e.g., phosphorus), ILs possess supramolecular structure. So they can affect the crystallization and growth of Ni-based catalysts, resulting in their special structures and morphologies with high performance. (2) ILs act as a media, templates and reactants simultaneously, which are expected to simplify the preparation and post-treatment process of catalysts.

## 3. Using Deep Eutectic Solvents to Prepare Nickel-Based Electrocatalysts

As analogues of ILs, DESs have the similar advantages to ILs in the synthesis of nanomaterials. DESs can also be used as media, templates and reactants to obtain Ni-based materials with highly electrochemical performance. We summarize the relevant literatures in Table 2.

### 3.1. Deep Eutectic Solvents as Solvents and Templates to Prepare Nickel-Based Electrocatalysts

In DESs, a conventional NGR can also occur, just like in aqueous solution. That is, a metal with high activity replaces a metal with low activity in DESs. For example, in ChCl/EG containing $Ni^{2+}$, self-supporting Ni-Fe bimetallic nanoparticles could be obtained by NGR through using Fe as sacrificial template. This compound had wonderful OER performance with $\eta_{10}$ of 319 mV [52].

The reactivity series of metals are affected by some factors, such as solvents, pH, and precipitants. As expected, DESs have ability to change the standard redox potential values of metals since the liquid structure of DESs is different from that of water. Different environments of DESs and $H_2O$ lead to different degrees of difficulty for atoms to lose outer electrons, that is, different standard electrode potentials. This property of DESs is attributed to the fact that DESs can form complexes with metal salts. It is well known that Cu and $Ni^{2+}$ can't occur NGR without reducing agent in the aqueous solution. However, in ChCl/EG, the redox potentials of $Cu^+/Cu$ and $Ni^{2+}/Ni$ are $-0.350$ V and $-0.154$ V at 353 K, respectively (Figure 4a). Therefore, Cu and $Ni^{2+}$ can undergo NGR in ChCl/EG. Zhang's group used this property to load Ni nanostructures on Cu foils successfully [53]. Adding the Cu foils into ChCl/EG-$NiCl_2$ induced the substitution reaction between Cu and $Ni^{2+}$. Cu converted into $CuCl_3^{2-}$ through releasing electrons, which was accompanied with cracking on the Cu foils, resulting in rough and porous surfaces. With the reaction proceeding, $Ni^{2+}$ was gradually reduced to Ni nanocrystals, and deposited on the Cu

foils. A self-supporting 3D nanoporous Ni films were finally generated (Figure 4b–d). The 3D Ni nanocrystallites@Cu exhibited high performance for HER, which the overpotential required only 170 mV to get 10 mA cm$^{-2}$, while Tafel slope was 98.5 mV dec$^{-1}$ (Figure 4e). Based on the results of this study, Zhang's research group further added a sulfur source to the ChCl/EG-NiCl$_2$ system [54]. When the NGR occurred between the Cu foils and Ni$^{2+}$, S was doped into the Ni films, obtaining S-doped Ni microsphere films on Cu foils (Ni$_3$S$_2$@Cu), as shown in Figure 4f,g. The doping of heteroatom could adjust the electronic structure of nickel, thereby optimizing the free energy of H$_2$ adsorption (Figure 4h–j). Ni$_3$S$_2$@Cu showed high catalytic activity and stability for HER in a wide pH range through in situ activation. The overpotential required only 91.6 mV and 60.8 mV to get 10 mA cm$^{-2}$ in 0.5 M H$_2$SO$_4$ and 1.0 M KOH, respectively, better than the Ni nanocrystallites@Cu mentioned above. The effect of DESs on metal activity enables some reactions that cannot be carried out in the aqueous phase to proceed smoothly. This strategy would inspire researchers to use similar methods to synthesize a series of nickel-based catalysts, such as nickel phosphides, nickel selenides, etc., in order to screen more excellent catalysts.

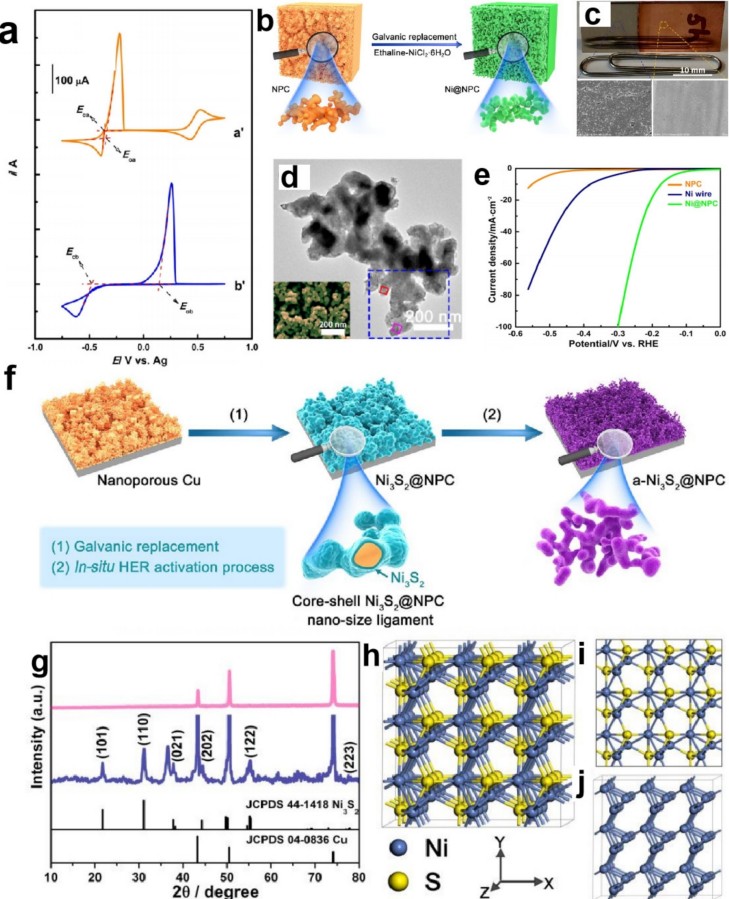

**Figure 4.** CVs of 0.10 M CuCl$_2$·2H$_2$O (a') and 0.10 M NiCl$_2$·6H$_2$O (b') in ChCl/EG recorded at Pt electrode, respectively (**a**), schematic diagram for the synthesis of Ni nanocrystals (**b**), photographic image for Cu foil before and after replacement for 5 h in 0.10 M NiCl$_2$·6H$_2$O/DES and corresponding SEM analysis (**c**), HAADF-STEM image (**d**) of Ni nanocrystals, polarization curves (**e**) of Ni nanocrystals and comparative electrocatalysts for HER (Reprinted with permission from Ref. [53], 2016, Elsevier); Schematic illustration showing the fabrication process of Ni$_3$S$_2$@Cu (**f**), XRD patterns (**g**) of Ni$_3$S$_2$@Cu (navy line) and bare Cu (pink line), front and top view of Ni$_3$S$_2$ supercell (3 × 3 × 3) shows the crystal structure (**h**,**i**), network of Ni-Ni bond paths in heazlewoodite Ni$_3$S$_2$ crystallizes (**j**) (Reprinted with permission from Ref. [54], 2017, Elsevier).

**Table 2.** Summary of HER, OER and Water Splitting Performance of Catalysts Involved in the Section 3.

| Catalyst | Morphology | Applied DES/IL | Synthesis Method | Synthesis Condition | Catalytic Performance | | | | | | | | Ref. |
|---|---|---|---|---|---|---|---|---|---|---|---|---|---|
| | | | | | HER | | | OER | | | Water Splitting | | |
| | | | | | Electrolyte | $\eta$ (mV)@ Current Density (mA cm$^{-2}$) | Tafel Slope (mV dec$^{-1}$) | Electrolyte | $\eta$(mV)@ Current Density (mA cm$^{-2}$) | Tafel Slope (mV dec$^{-1}$) | Electrolyte | Potential (V)@ Current Density (mA cm$^{-2}$) | |
| NiFe | Nanoparticles | ChCl/EG | Galvanic replacement reaction | 80 °C, 5 h | – | – | | 1 M KOH | 319@10 | 41.2 | – | – | [52] |
| Ni | Nanoparticles | ChCl/EG | Galvanic replacement reaction | 80 °C, 5 h | 1 M KOH | 170@10 | 98.5 | – | – | – | – | – | [53] |
| Ni$_3$S$_2$ | 3D hierarchically porous morphology | ChCl/EG | Galvanic replacement reaction | 80 °C, 5 h | 1 M KOH | 60.8@10 | 67.5 | – | – | – | – | – | [54] |
| Ni$_3$S$_2$ | 3D hierarchically porous morphology | ChCl/EG | Galvanic replacement reaction | 80 °C, 5 h | 0.5 M H$_2$SO$_4$ | 63.5@10 | 91.6 | – | – | – | – | – | [54] |
| Ni | Cubic morphology | ChCl/EG | Electrodeposition | −0.98 V (vs. Ag/Ag$^+$), 900 s, 45 °C | 1 M KOH | 154@10 | 147 | – | – | – | – | – | [55] |
| Metals | – | Choline-based ILs | Selective extraction | – | – | – | – | – | – | – | – | – | [56] |
| Ni/TiO$_2$ | Spherical morphology | ChCl/EG | Electrodeposition | 5–15 mA cm$^{-2}$, 40 °C | 1 M NaOH | – | 122 | – | – | – | – | – | [57] |
| Ni/Ni(OH)$_2$ | Irregular morphology | ChCl/EG | Electrodeposition | −0.85 V (vs. Ag/Ag$^+$), 70 °C | 1 M KOH | 110@10 | 83.9 | 1 M KOH | 290@10 | 120.9 | – | – | [58] |
| Co$_x$-Ni(OH)$_2$ | Ultrafine nanoparticles | ChCl/EG | Electrodeposition | −0.3 V to −1.1 V (vs. Ag/Ag$^+$), 60 °C | 1 M KOH | 106@10 | 98.2 | 1 M KOH | 330@100 | 126.7 | – | – | [59] |
| NiS$_x$ | Cauliflflower-like morphology | ChCl/EG | Electrodeposition | −0.65 V (vs. Ag/Ag$^+$), Room temperature | 1 M KOH | 54@10 | 54 | – | – | – | – | – | [60] |
| Ni$_2$P | Nanorods | [BMIM] Br | Ionothermal and calcining methods | 150 °C, 72 h; 700 °C, 2 h | – | – | – | – | – | – | – | – | [61] |
| NiP$_x$ | Nanoparticles | ChCl/EG | Electrodeposition | 0.54 V (vs. Ag/Ag$^+$), Room temperature | 1 M KOH | 105@10 | 44.7 | – | – | – | – | – | [62] |
| Ni-Cu | Nanosheets | ChCl/EG | Electrodeposition | −0.55 V to -0.90 V (vs. Ag/Ag$^+$), 60 °C | 1 M KOH | 128@10 | 57.2 | – | – | – | – | – | [63] |
| Ni-Co-Sn | Agglomerated nodular-grainsmorphology | ChCl/EG | Electrodeposition | 30 mA cm$^{-2}$, Room temperature | 1 M KOH | – | 121 | – | – | – | – | – | [64] |
| Ni-Fe | Nanoparticles | ChCl/EG | Electrodeposition | −0.9 V (vs. Ag/Ag$^+$), 60 °C | 0.1 M KOH | 316@10 | 62 | – | – | – | – | – | [65] |

**Table 2.** *Cont.*

| Catalyst | Morphology | Applied DES/IL | Synthesis Method | Synthesis Condition | HER | | | OER | | | Water Splitting | | Ref. |
|---|---|---|---|---|---|---|---|---|---|---|---|---|---|
| | | | | | Electrolyte | $\eta$ (mV)@ Current Density (mA cm$^{-2}$) | Tafel Slope (mV dec$^{-1}$) | Electrolyte | $\eta$(mV)@ Current Density (mA cm$^{-2}$) | Tafel Slope (mV dec$^{-1}$) | Electrolyte | Potential (V)@ Current Density (mA cm$^{-2}$) | |
| Ni-Mo-Cu | Irregular block-shaped particles | ChCl/urea | Electrodeposition | −1.2 V (vs. Ag/Ag$^+$), 70 °C | 1 M KOH | 93@10 | 105 | – | – | – | – | – | [66] |
| Ni-Mo | Microsphere | ChCl/EG | Electrodeposition | −0.95 V (vs. Ag/Ag$^+$), 60 °C | 1 M KOH | 63@20 | 49 | 1 M KOH | 335@20 | 108 | 1 M KOH | 1.59@10 | [67] |
| S-NiFe$_2$O$_4$/Ni$_3$Fe | Porous networks | ChCl/EG | Electrodeposition | −1.0 V (vs. Ag/Ag$^+$), 60 °C | – | – | – | 1 M KOH | 260@10 | 35 | 1 M KOH | 1.52@10 | [68] |
| NiCo$_x$S$_y$ | Microsphere arrays | ChCl/EG | Electrodeposition | −0.80 V (vs. Ag/Ag$^+$), 60 °C | 1 M KOH | 65@20 | 54 | 1 M KOH | 270@20 | 35 | 1 M KOH | 1.57@10 | [69] |
| (FeCoNiCuZn) (C$_2$O$_4$)· 2H$_2$O | 2D morphology | PEG/oxalic acid | Ionothermal method | 160 °C, 8 h | – | – | – | 1 M KOH | 334@10 | 67.93 | – | – | [70] |
| NiCo$_2$O$_4$@NiMoO$_4$ | Nanorods | – | Hydrothermal method | 120 °C, 10 h; 80 °C, 6 h | 1 M KOH | 170@10 | 184.25 | 1 M KOH | 300@10 | 94.06 | 1 M KOH | 1.65@10 | [71] |
| Ni$_{0.33}$Co$_{0.67}$S$_2$ | Nanowires | – | Hydrothermal and calcining methods | 90 °C, 8 h; 250 °C 2 h | 0.5 M H$_2$SO$_4$ | 73@10 | 44.1 | – | – | – | – | – | [72] |
| NiCo$_2$O$_4$ | Nanooctahedron | ChCl/glycerol | Calcining method | 500 °C, 15 min, air atmosphere | – | – | – | 1 M KOH | 320@10 | 67 | – | – | [73] |
| NiCo$_2$S$_4$ | Spherical sea urchin-like nanostructures | PEG 200/thiourea | Ionothermal method | 160.15 °C, 16 h | – | – | – | 1 M KOH | 337@10 | 64 | – | – | [74] |
| Fe$_3$S$_4$ | Nanosheets | PEG 200/thiourea | Ionothermal method | 200 °C, 16 h | – | – | – | – | – | – | – | – | [75] |
| NiS/graphene | 2 D morphology | NiCl$_2$· 6H$_2$O/PEG 200 | Calcining method | 550 °C, 4 h, N$_2$ atmosphere | 1 M KOH | 70@10 | 50.1 | 1 M KOH | 300@10 | 55.8 | 1 M KOH | 1.54@10 | [76] |
| NiS$_2$/ graphene | Nanosphere | NiCl$_2$· 6H$_2$O/malonic acid | Calcining method | 550 °C, 4 h, N$_2$ atmosphere | 1 M KOH | 57@10 | 47 | 1 M KOH | 294@10 | 54 | 1 M KOH | 1.52@10 | [77] |
| Ni$_2$P/graphene | Nanoparticles | NiCl$_2$· 6H$_2$O/malonic acid | Calcining method | 400 °C, 4 h, N$_2$ atmosphere | 1 M KOH | 103@10 | 56.5 | 1 M KOH | 275@20 | 56.2 | 1 M KOH | 1.51@10 | [78] |
| N-C/NiS$_2$ | Sheet-like 2D nanostructures | NiCl$_2$· 6H$_2$O/urea | Calcining method | 550 °C, 4 h, N$_2$ atmosphere | 1 M KOH | 78@10 | 63.4 | 1 M KOH | 264@10 | 51.3 | 1 M KOH | 1.53@10 | [79] |
| NiFe-LDH/N-C | flflower-like structure | NiCl$_2$· 6H$_2$O/ FeCl$_3$· 6H$_2$O/urea/water | Ionothermal method | 120 °C, 12 h | – | – | – | 0.1 M KOH | 363@500 | 49.8 | – | – | [80] |

**Table 2.** *Cont.*

| Catalyst | Morphology | Applied DES/IL | Synthesis Method | Synthesis Condition | Catalytic Performance | | | | | | | | Ref. |
|---|---|---|---|---|---|---|---|---|---|---|---|---|---|
| | | | | | HER | | | OER | | | Water Splitting | | |
| | | | | | Electrolyte | $\eta$ (mV)@ Current Density (mA cm$^{-2}$) | Tafel Slope (mV dec$^{-1}$) | Electrolyte | $\eta$(mV)@ Current Density (mA cm$^{-2}$) | Tafel Slope (mV dec$^{-1}$) | Electrolyte | Potential (V)@ Current Density (mA cm$^{-2}$) | |
| NiFe-LDH | 2D morphology | FeCl$_3\cdot$6H$_2$O/urea | Dipping-redox method | 60 °C, 30 s | 1 M KOH | 160@10 | 42 | 1 M KOH | – | – | 1 M KOH | 1.61@10 | [81] |
| N,S-FeCoNi | Oriented-grow flower | FeCl$_3\cdot$6H$_2$O/ CoCl$_2\cdot$6H$_2$O/ NiCl$_2\cdot$6H$_2$O/L-cysteine | Calcining method | 350 °C, 12 h, N$_2$ atmosphere | – | – | – | 1 M KOH | 251@10 | 58 | – | – | [82] |
| High-entropy metal phosphides | Nanoparticles | [P$_{4444}$]Cl/ethylene glycol /five equimolar hydrated metal chlorides | Eutectic solvent method | in an inert atmosphere at 400 °C for 3 h | 1 M KOH | 136@10 | 85.5 | 1 M KOH | 320@10 | 60.8 | 1 M KOH | 1.78@100 | [83] |

Like ILs, DESs are excellent electrolytes due to unique physical and chemical properties, such as large electrochemical window. The researchers compared the morphologies and catalytic performance of Ni nanoparticles prepared by electrodeposition in acetate buffer and ChCl/EG DES electrolytes, respectively [55]. By comparison, it was found that the electrodeposition process in DES was more efficient than that of acetate buffer, producing Ni nanoparticles with a more uniform needle-shaped structure and smaller size. Therefore, DES-mediated Ni nanoparticles exhibited better electrocatalytic performance, which required overpotentials both HER and OER of 154 mV and 350 mV to achieve 10 mA cm$^{-2}$, respectively with smaller Tafel slopes. In contrast, Ni from acetate required overpotentials of 164 mV and 400 mV for HER and OER.

Further studies found that the type of DESs and metal salts also affect the compositions and morphologies of the products in the process of electrodeposition, which in turn influence performance of water splitting. Both ChCl/EG and ChCl/urea have been tested to use as electrolytes to electrolyze nickel salts. ChCl/EG has higher conductivity and lower viscosity than ChCl/urea, making the diffusion rate of Ni$^{2+}$ higher than that in ChCl/urea. Meanwhile, the coordination environment and interactions between nickel and ChCl/EG or ChCl/urea are also different. Therefore, two distinct morphologies of Ni films were obtained from the two solutions, and exhibited different HER performance [56,57].

In the process of preparing nickel-based catalysts, NiCl$_2$ and Ni(NO$_3$)$_2$ are commonly used. Different discharge capacity of anions leads to different products. In the ChCl/EG-NiCl$_2$ system, the result product was mainly Ni films by electrodeposition technology. However, the addition of Ni(NO$_3$)$_2$ to the ChCl/EG-NiCl$_2$ system would alter the product from pure Ni films to Ni/Ni(OH)$_2$ films. The reason was that NO$_3^-$ generated OH$^-$ at the reduction potential, inducing the formation of Ni(OH)$_2$ on the surface of Ni films. To get 10 mA cm$^{-2}$, Ni/Ni(OH)$_2$ required the overpotential of 110 mV for the HER, while 320 mV for the OER [58]. Using this property of nitrate, adding Ni(NO$_3$)$_2$ and Co(NO$_3$)$_2$ to ChCl/EG obtained Co-Ni(OH)$_2$, a multifunctional catalyst for HER, OER, water splitting, and urea oxidation reaction [59].

In ChCl/EG DES containing NiCl$_2$·6H$_2$O, adding different raw materials can easily realize the controllable design of catalysts. Adding S or P source during the electrodeposition process could get S-doped Ni films [60] or P-doped Ni films [61,62], respectively. For example, S-doped Ni microsphere films caused a large number of oxygen vacancies on the surface and change the morphology of catalyst. Catalytic performance varied greatly with different Ni/S ratio. It was found that the optimal ratio of Ni and S was 1:0.25. NiS$_{0.25}$ possessed the biggest surface area compared with other ratios and exhibited optimal HER performance. The $\eta_{10}$ was only 54 mV and continuously catalyzed for 60 h in KOH.

In addition, a variety of Ni-based multimetallic catalysts are also prepared using DESs as media, such as Ni-Cu [63], Ni-Co-Sn [64], Ni-Fe [65], Ni-Mo-Cu [66], Ni-Mo [67], S-NiFe$_2$O$_4$/Ni$_3$Fe [68], NiCo$_x$S$_y$ [69], (FeCoNiCuZn) (C$_2$O$_4$)·2H$_2$O [70], etc. Vo et al. proposed a facile and scalable electrodeposition strategy to directly grow self-supporting Ni-Fe alloy thin films on the surface of copper foils by simultaneously adding NiCl$_2$·6H$_2$O and FeCl$_3$·6H$_2$O to ChCl/EG DES [65]. DES has a wide potential window and the ability to suppress side reactions, such as in situ the precipitation of Fe(OH)$_3$ (by-product) on the electrode surface which often occurs in aqueous solution. Moreover, Ni-Fe co-deposition could be easily obtained from DES without adding complexing agent which was indispensably used in aqueous solution. In addition, DES made the deposited Ni-Fe layer uniform and dense. Clearly, the electrodeposition in DES is better than that in aqueous phase. By screening the ratio of Ni/Fe in DES solution, Ni$_{0.75}$Fe$_{0.25}$ was found to exhibit the best catalytic performance for OER. Only a 316 mV overpotential was required to achieve a current density of 10 mA cm$^{-2}$, while the Tafel slope was 62 mV dec$^{-1}$. It indicated that DES-derived Ni-Fe alloy was a high-performance and stable electrocatalyst for water splitting. Yang et al. applied DESs (polyethylene glycol-critic acid (PEG-CA) DES, PEG-oxalic acid (PEG-OA) DES, and EG-CA DES to synthesize multi-metal oxalate consisting of Fe, Co, Ni, Cu, and Zn [70]. It was found that the structure and composition of DESs could

control the nucleation and growth of multi-metal oxalate. High molecular PEG resulted to the gradually slow-release oxalic acid. So, PEG-CA promoted the formation of a 2D structure. The low molecular EG-OA DES induced the formation of 3D nanoparticles. Notably, owing to its different active site exposure, layered structure, stronger electron withdrawing ability and electron coordination ability of oxalate anion, the well-designed 2D (FeCoNiCuZn) $(C_2O_4) \cdot 2H_2O$ nanosheets as electrocatalyst exhibited an overpotential of 334 mV at 100 mA cm$^{-2}$ for OER.

The other great advantage of DESs is their ability to change the reaction mechanism. Generally, the preparation of metal oxides generally goes through two-step method. The first step is to synthesize the precursor, such as oxalate, carbonate, hydroxide and various inorganic salt. The second step is to convert precursor into metal oxide through calcination method in air or oxygen atmosphere [71,72]. Both steps usually require a long reaction time, harsh reaction conditions and complicated operation procedures, etc. The intervention of DESs has ability to successfully solve the above problems. At room temperature, a hydrolyzable amine-contained Ni-Co complex with regular octahedron shape was synthesized using ChCl/glycerol DES as a medium. DES could inhibit the hydrolysis of this complex in the absence of hydrolysis inhibitors, avoiding the formation of nickel and cobalt hydroxides. Calcining this precursor in air, the $NiCo_2O_4$ with octahedral structure was gotten in just 15 min. The unique octahedral morphology, good dispersion, and the synergistic effect between Ni and Co ions endowed the obtained $NiCo_2O_4$ with excellent electrochemical oxygen evolution performance and stability. In 1.0 M KOH, the $\eta_{10}$ and Tafel slopes were 320 mV and 67 mV dec$^{-1}$, respectively [73].

### 3.2. Deep Eutectic Solvents as Reaction Reagents to Prepare Nickel-Based Electrocatalysts

The designability of DESs motivates researchers to design novel DESs as metal sources, carbon sources and other active ingredients (such as P, S, etc.) through exploring suitable HBDs or HBAs. Mu research group cleverly designed a new type of polyethylene glycol (PEG) 200/thiourea DES [74,75]. Besides acting as a solvent and template, this DES could simultaneously use as a sulfur source to directly prepare sulfides through one-step reaction with $NiCl_2$ and $CoCl_2$. The obtained $NiCo_2S_4$ had excellent OER performance in 1.0 M KOH ($\eta_{10}$ = 337 mV, Tafel slope = 64 mV dec$^{-1}$). The catalytic performance of $NiCo_2S_4$ remained almost unchanged after continuous cycles of 2000 cycles [74]. Further, using this PEG-based DES, a series of metal sulfides were prepared, including $NiS_2$, $CoS_2$, $NiCo_2S_4$, $CoS_2$ + $Ni_2CoS_4$ [74] and $Fe_3S_4$ [75]. The difference between the catalysts obtained by DES-mediated method and the conventional approach was studied in detail. There existed synergistic effects in DES-mediated catalysts, including high surface area, rich active sites, smooth diffusion of electrolytes and $O_2$, as well as strong structural integrity. All these features contributed to the high activity and long-term stability of OER catalysts. This study offered a new synthetic approach for metal sulfides with hierarchically structure.

The above achievement has been realized in the preparation of electrocatalysts by using one component of DESs. However, this strategy still wastes another component of DESs. Today, zero emission is beautiful vision to protect the environment, and people are actively working towards this goal. One of the important measures is to make all components of DESs to participate in reaction through one-step preparation strategy, which is called "all-in-one" strategy. This not only improves the utilization rate of raw materials, but also conforms to the theory of atomic economics. In this research field, DESs based on hydrated metal chlorides have attracted much attention. It is proved that a variety of DESs can be obtained through combining $NiCl_2 \cdot 6H_2O$ with carbon-rich HBDs, such as PEG 200, malonic acid, tartaric acid, citric acid, glucose, fructose, etc. [76–78]. Different liquid structures, reducibility, acidity and alkalinity of carbon-rich materials will result in different nucleation and growth mechanisms in preparing catalysts. So $NiCl_2 \cdot 6H_2O$/carbon-rich HBDs offer a series of carbon-contained nickel-based catalysts with different structures and properties. Addition of carbon into catalysts can solve the problem of their intrinsic

low electronic conductivity, thereby effectively improving the catalytic activity. This idea provides an effective strategy for green and suitable ideal catalysts.

Further, in $NiCl_2 \cdot 6H_2O$/carbon-rich carboxylic acid DESs, carbon-coated nickel sulfides, nickel phosphides are controllably synthesized by adding corresponding heteroatom sources. For example, the 2D NiS/graphene heterostructure was gotten by calcining $NiCl_2 \cdot 6H_2O$/PEG 200 DES with sublimation sulfur [76,77]. Changing the ratio of PEG 200 in DES would adjust the content of graphene of catalyst, which easily tuned the catalytic performance. The catalyst with the best ratio of graphene had excellent HER and OER performance under alkaline conditions, while it showed good property of overall water splitting (a low cell voltage of 1.54 V at 10 mA cm$^{-2}$). Similarly, adding $NaH_2PO_2 \cdot 2H_2O$ to the $NiCl_2 \cdot 6H_2O$/malonic acid DES could produce $Ni_2P$@graphene, which required a voltage of 1.51 V to reach 10 mA cm$^{-2}$ for water splitting [78].

Urea can also act as HBD to form DESs with hydrated metal chlorides, which provides N and C elements to catalysts. For example, sulfurizing $NiCl_2 \cdot 6H_2O$/urea DES could synthesize two-dimensional N-C/$NiS_2$ nanosheets, which had synergistic effect between $NiS_2$ and N-C (Figure 5a–g) [79]. The obtained N-C/$NiS_2$ nanosheets showed excellent performance of water splitting, requiring 1.53 V to undertake 10 mA cm$^{-2}$.

More importantly, hydrated metal chlorides/urea DESs not only simultaneously introduce N and C elements into catalysts, but also provide metal elements and OH$^-$. When pyrolyzing $NiCl_2 \cdot 6H_2O$/$FeCl_3 \cdot 6H_2O$/urea/water DES, all components contributed to the structure of catalyst, resulting in the formation of NiFe LDH ultrathin nanosheets hybridized with N-doped carbon quantum dots [80]. Among them, the formation of LDH was due to the ability of urea to hydrolyze OH$^-$ in the water environment. In situ carbonization of DES produced N-doped carbon quantum dots, which made up for the deficiency of carbon quantum dots needed to prepare in advance among other preparation methods. The catalyst exhibited excellent OER performance. The overpotentials were 252 mV, 311 mV, and 363 mV to achieve 10 mA cm$^{-2}$, 100 mA cm$^{-2}$, and 500 mA cm$^{-2}$, respectively.

However, the NiFe-LDH obtained in above way need to be spread on the substrate surface when they are used as catalysts. The interfacial space between catalyst and electrode substrate will hinder the electron transport in some degree. To address this limitation, our research group directly immersed NF into $FeCl_3 \cdot 6H_2O$/urea DES, in which NF served as the electrode substrate as well as Ni source [81]. The nano-flower-like NiFe-LDH was obtained on the surface of NF in a very short time. A series of reactions occurred when NF was immersed in $FeCl_3 \cdot 6H_2O$/urea. $Fe^{3+}$ reacted with NF, and NF was oxidized to $Ni^{2+}$. Then, there was reaction among $Fe^{3+}$ and $Ni^{2+}$ and OH$^-$ produced by the hydrolysis of urea formed NiFe-LDH on the surface of NF. With no interfacial gap, excellent uniformity of catalyst morphology synergistically resulted in excellent electrocatalytic activity. As cathode and anode, the as-prepared NiFe-LDH could implement very well toward water splitting and urea electrolysis, a hydrogen production reaction to replace water splitting.

In order to further improve the catalytic activity of catalysts, an integrated DESs strategy is adopted to introduce multiple metals and heteroatoms into the catalyst simultaneously. Three hydrated metal halides, including $NiCl_2 \cdot 6H_2O$, $FeCl_3 \cdot 6H_2O$ and $CoCl_2 \cdot 6H_2O$, were used as HBAs, while L-cysteine containing -COOH and -$NH_2$ functional groups was employed as HBD [82]. Mixing them by a certain molar ratio would obtain a new type of hydrated trimetallic salt/L-cysteine DES. Calcining this DES directly fabricated N,S-FeCoNi (Figure 5h–j). In this reaction, DES had many functions, including solvent, template agent, N source, S source, and metal sources. The hydrogen bond network structure of DES made three metals (Fe, Co and Ni) as well as binary heteroatoms (N and S) uniformly distribute in the resulting product. The prepared catalyst had unique structural advantages, such as high surface area, porous structure, and evenly dispersed metal ions. Fe, Co, Ni, N, and S cooperated with each other to adjust the electronic structure, which was beneficial to charge transport and improved electrocatalytic performance. An overpotential was required 251 mV to get 10 mA cm$^{-2}$ in 1.0 M KOH.

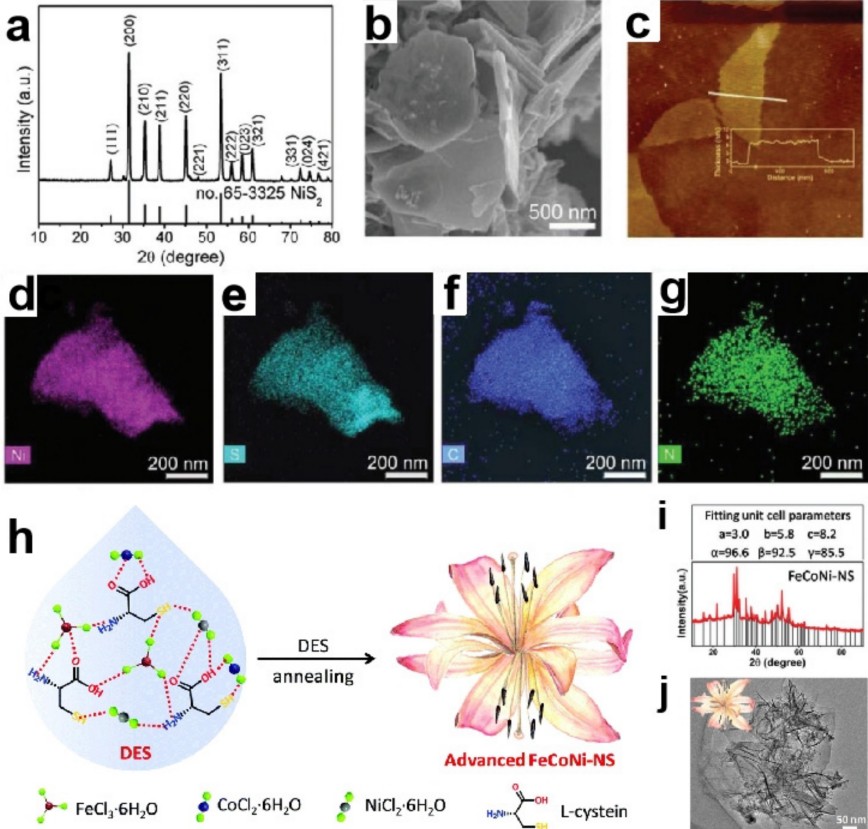

**Figure 5.** XRD pattern (**a**), SEM image (**b**), AFM image (**c**) and elemental mapping images (**d**–**g**) of the N–C/NiS$_2$ nanostructures. (Reprinted with permission from Ref. [79], 2020, Royal Society of Chemistry); Schematic representation of the formation of N,S-FeCoNi from the DES (**h**), XRD pattern (**i**) and TEM image (**j**) of N,S-FeCoNi (Reprinted with permission from Ref. [82], 2019, Royal Society of Chemistry).

Mu research group designed a new type of DES through mixing five equimolar hydrated metal chlorides (FeCl$_3$·6H$_2$O, CoCl$_2$·6H$_2$O, NiCl$_2$·6H$_2$O, MnCl$_2$·4H$_2$O and CrCl$_3$·6H$_2$O) and [P$_{4444}$]Cl into EG simultaneously [83]. Among them, hydrated metal chloride and [P$_{4444}$]Cl were HBAs, while EG was HBD. Phosphating this DES in situ in an inert atmosphere obtained high-entropy metal phosphide (HEMP), which contained a variety of elements and the content of each element was between 5–35%. A variety of active groups in HBD are simultaneously combined with hydrated metal chlorides, leading to the uniform dispersion of different metals in the prepared catalyst. This will facilitate different metals to form synergistic systems and improve catalytic efficiency. As a dual-function catalyst, it presented $\eta_{10}$ of 136 mV for HER as well as $\eta_{10}$ of 320 mV for OER in 1.0 M KOH. The catalytic performance of HEMP was better than a single component and IrO$_2$ catalyst. From the above design of multimetallic catalysts, it can be seen that the design of HBDs is critical to excellent catalysts.

The "all-in-one" strategy for the preparation of heteroatom-doped nickel-based catalysts through designing DESs composed with hydrated metal halides as HBAs and heteroatom sources as HBDs is shown in Figure 6.

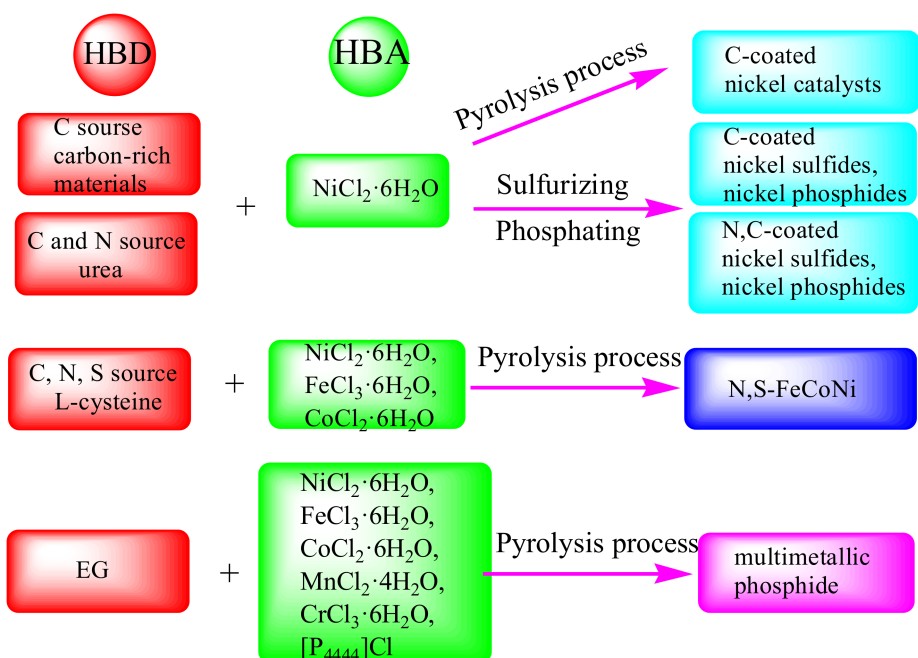

**Figure 6.** Schematic diagram of the "all-in-one" strategy to prepare heteroatom-doped nickel-based catalysts through designing composition of DESs [76–80,82,83].

## 4. Summary and Outlook

It can be seen that using ILs and DESs to synthesize nickel-based water splitting electrocatalysts has received widespread attention. This IL/DES-mediated preparation method has the following advantages: (1) ILs and DESs can act as media, templates and reactants, which not only control the morphologies and structures of the catalysts but also make the reaction system simple. Meanwhile, ILs and DESs can decrease emission, further reducing environmental pollution. This strategy can improve the atomic conversion rate of ILs and DESs, in line with the theory of atomic economics. (2) ILs and DESs may affect the nucleation and crystallization of catalysts, resulting in the synthesis of novel materials with wonderful performance. (3) The compositions and structures of ILs and DESs could be rationally designed according to the requirement of catalysts. The designability of ILs (diversity of cations and anions) and DESs (multiple of HBDs and HBAs) will give novel ILs and DESs, providing more opportunities for the preparation of novel catalysts. To help the researchers gain basic knowledge on the ILs and DESs method, we summarize the advantages of ILs and DESs in different synthetic methods and compare them with traditional methods, as shown in Table 3.

However, there are still some urgent problems to be solved. First, the controllable preparation of nickel-based catalysts based on ILs and DESs is an urgent problem to be solved. One of the ultimate goals of catalyst research is to achieve a controllable study among the composition, structure and catalytic performance. In order to achieve this goal, researchers have been working on this research. Wegner et al. have reviewed in detail the interdependencies of ILs with the metal nanoparticles and tried to elucidate influences of the IL-cation, -anion and alkyl chain length [84]. However, theoretically guided research on the effects of ILs and DESs on inorganic nanomaterials is currently in infancy. Therefore, the preparation of catalysts using ILs and DESs requires further theoretical studies, including the nucleation and growth processes of metal catalysts induced by them, which will be the theoretical basis to realize controllable preparation of catalysts.

**Table 3.** Comparative of the advantages of IL/DES synthesis over traditional methods.

| Method | Limitations of Traditional Synthesis | Advantages of IL/DES Synthesis |
|---|---|---|
| Precipitation method | The products have poor dispersibility, being aggregate easily. | 1. ILs and DESs are able to dissolve some substances that are difficult to dissolve in conventional solvents;<br>2. ILs and DESs stabilize nanoparticles by electrostatic force and other forces, so that nanoparticles have good dispersibility;<br>3. ILs and DESs can act as templating agents to change the morphology of products. |
| Sol-gel method | 1. The aging time is difficult to control;<br>2. The pore structure is prone collapse or shrinkage during the process from sol-to-gel and product drying. | 1. ILs and DESs have almost no vapor pressure, resulting in a longer aging time from sol to gel, which is conducive to the formation of a stable gel structure;<br>2. ILs and DESs can avoid the collapse or shrinkage during the process from sol to gel and product drying;<br>3. ILs and DESs enable better control of specific surface area and pore structure. |
| Solvothermal/hydrothermal method | 1. The reaction conditions are harsh and the equipment requirements are high;<br>2. The security is poor;<br>3. Some substances are difficult to dissolve, such as metals, metallic oxides, etc. | 1. ILs and DESs are almost nonvolatile and have highly thermal stability. Therefore, they can maintain normal pressure or even vacuum at high temperature in reactions, which not only reduce the production of toxic gases but also avoid the danger of high-pressure reactions;<br>2. ILs and DESs can dissolve materials that are difficult to dissolve in conventional solvents, such as metals, metallic oxides and so on, so that the reactions can proceed homogeneously;<br>3. ILs and DESs have special physicochemical properties and structures. On the one hand, ILs and DESs have greater viscosity, higher polarity, lower surface tension than aqueous solution or conventional organic solvents. On the other hand, they can spontaneously form extended hydrogen bonds. Therefore, ILs and DESs can regulate the morphology of inorganic nanomaterials, while tune the nucleation and growth of catalysts. |
| Electrodeposition | 1. The aqueous electrolyte has narrow electrochemical window;<br>2. Hydrogen is generally generated in the aqueous electrolyte to interfere with the quality of the precipitate;<br>3. Organic electrolyte system is volatile and limited by temperature;<br>4. The high temperature molten salt system is highly corrosive. | 1. The liquid ranges of ILs and DESs is wide, which are from below or close to room temperature to above 300 °C with high thermal and chemical stability;<br>2. ILs and DESs have small vapor pressure, making storage and use process be simple;<br>3. ILs and DESs have high conductivity and large electrochemical window; |

Second, an "all-in-one" synthesis strategy needs to be to further studied. The biggest challenge concerning this strategy is how to creatively select ILs and DESs to obtain target products with excellent performance. This method brings practical benefits to the improvement of technology and cost saving, helping people achieve carbon peaking and carbon neutralization.

**Author Contributions:** C.Z.: Literature collection and manuscript writing; T.B.: Literature collection and analysis; Y.S.: Content design; B.X.: Content design, revision, project administration and funding acquisition; S.Z.: Literature collection. All authors have read and agreed to the published version of the manuscript.

**Funding:** This work was financially supported by the Natural Science Foundation of Shandong Province (No. ZR2015EM022), Shandong Provincial Department of Education 2020 undergraduate teaching reform research project (No. M2020125), Experimental technology project of Dezhou University (No. SYJS20007), National College Student Innovation Training (No. S202110448037, X202110448016 and X202110448042).

**Conflicts of Interest:** The authors declare no conflict of interest.

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
