# Peer review of "Ionic Liquid/Deep Eutectic Solvent-Mediated Ni-Based Catalysts and Their Application in Water Splitting Electrocatalysis"

_catalysts, doi:10.3390/catal12080928_

Round 1
Reviewer 1 Report
The work presented to the review, entitled „Ionic liquid/deep eutectic solvent-mediated Ni-based catalysts and their application in water splitting”, is very interesting. I appreciate this work and would like to recommend it for publication in Catalysts after minor corrections.
My suggestions and comments are as follows:
The sentence presented in the article „ILs are composed of organic cations and inorganic/organic anions” is incorrect. In the literature [Shobukawa H., Tokuda H., Tabaata S., Watanabe M., Electrochim. Acta, 50, 305 (2004).] ionic liquids with inorganic anions are described. Completely inorganic ionic liquids are also known [Dai L., Yu S., Shan Y., He M., Eur. J. Inorg. Chem., 237 (2004)].
In the sentence „Special compositions and structural features lead ILs to possess unique physical and chemical properties, such as low melting point and volatility, strong conductivity and so on.” Instead of „volatility” should be „nonvolatilty”, because ionic liquids do not evaporate.
The sentence: „Meanwhile, [EMIM]+ as a cationic surfactant…” is also incorrect. Firstly, the cation cannot be a surfactant, because the whole compound is the surfactant, secondly, 1-ethyl-3-methylimidazolium nitrate(V) is not a cationic surfactant. Just like not every sodium carboxylate is an anionic surfactant (e.g. sodium acetate is not whereas sodium laurate is), which of course is related to the appropriate length of the alkyl chain).
The sentence: „[BMIM]+ cation could interact with diethanolamine anion, which slowed down the diffusion rate of nickel cation and the decomposition rate of red phosphorus, increasing the porosity and pore size of as-prepared catalyst.” is incorrect. Diethanolamine cannot be an anion.
In the sentence: „Recently, our research group have explored the synthesis method of nickel phosphides based on quarter phosphonium salt [40]” instead of „quarter” should be „quaternary”.
Author Response
Aug. 12, 2022
Dr. Prof. Reviewer 1
Thank you very much for your letters on Thu., Aug 4, 2022, which were about our manuscript entitled “Ionic liquid/deep eutectic solvent-mediated Ni-based catalysts and their application in water splitting electrocatalysis” (catalysts-1856033). We appreciate you very much because all of the comments are very valuable and helpful for us to make our manuscript better. We have revised the manuscript on the basis of the comments and suggestions. The changes are highlighted in yellow color in the revised manuscript. A list of changes and answers to the comments are given in the following pages. In the list, we marked the page and line number to facilitate reviewers to review. However, it should be noted here that the automatically generated line number in the template is different on different computers. What we have marked may be different from the corresponding ones on your computer. We are very sorry for this. On behalf of the coauthors, I hope that you are satisfied with the changes and look forward to a possible further publication process. In case of any question, please do not hesitate to contact me. Thank you very much.
Sincerely yours,
Prof. Dr. Bingwei Xin
xinbingwei@dzu.edu.cn
Dezhou University
Response to the comments and changes to the manuscript
Comments and Suggestions for Authors
The work presented to the review, entitled „Ionic liquid/deep eutectic solvent-mediated Ni-based catalysts and their application in water splitting”, is very interesting. I appreciate this work and would like to recommend it for publication in Catalysts after minor corrections.
My suggestions and comments are as follows:
The sentence presented in the article „ILs are composed of organic cations and inorganic/organic anions” is incorrect. In the literature [Shobukawa H., Tokuda H., Tabaata S., Watanabe M., Electrochim. Acta, 50, 305 (2004).] ionic liquids with inorganic anions are described. Completely inorganic ionic liquids are also known [Dai L., Yu S., Shan Y., He M., Eur. J. Inorg. Chem., 237 (2004)].
In the sentence „Special compositions and structural features lead ILs to possess unique physical and chemical properties, such as low melting point and volatility, strong conductivity and so on.” Instead of „volatility” should be „nonvolatilty”, because ionic liquids do not evaporate.
The sentence: „Meanwhile, [EMIM]+ as a cationic surfactant…” is also incorrect. Firstly, the cation cannot be a surfactant, because the whole compound is the surfactant, secondly, 1-ethyl-3-methylimidazolium nitrate(V) is not a cationic surfactant. Just like not every sodium carboxylate is an anionic surfactant (e.g. sodium acetate is not whereas sodium laurate is), which of course is related to the appropriate length of the alkyl chain).
The sentence: „[BMIM]+ cation could interact with diethanolamine anion, which slowed down the diffusion rate of nickel cation and the decomposition rate of red phosphorus, increasing the porosity and pore size of as-prepared catalyst.” is incorrect. Diethanolamine cannot be an anion.
In the sentence: „Recently, our research group have explored the synthesis method of nickel phosphides based on quarter phosphonium salt [40]” instead of „quarter” should be „quaternary”.
Reply:
- About the problem for that “ILs are composed of organic cations and inorganic/organic anions” is incorrect.
We are very sorry for the error about the basic concept. In the revised manuscript, we change this sentence to “They are composed entirely of ions with relatively large volume differences, making them difficult to densely pack or crystallize.” according to the literatures [Shobukawa H., Tokuda H., Tabaata S., Watanabe M., Electrochim. Acta, 50, 305 (2004).; Dai L., Yu S., Shan Y., He M., Eur. J. Inorg. Chem., 237 (2004)]. We have added these two literatures in the revised manuscript as Ref. 12 and Ref. 13. Please see page 2 paragraph 2 line 2 (the automatically generated line 58).
- About the problem for that “Special compositions and structural features lead ILs to possess unique physical and chemical properties, such as low melting point and volatility, strong conductivity and so on.”
We are very sorry for this error. We have replaced “volatility” to “nonvolatilty”. Please see page 2 paragraph 2 line 5 (the automatically generated line 61).
- About the problem for that “Meanwhile, [EMIM]+ as a cationic surfactant…” is also incorrect. Firstly, the cation cannot be a surfactant, because the whole compound is the surfactant, secondly, 1-ethyl-3-methylimidazolium nitrate(V) is not a cationic surfactant. Just like not every sodium carboxylate is an anionic surfactant (e.g. sodium acetate is not whereas sodium laurate is), which of course is related to the appropriate length of the alkyl chain)
Thank you very much. We have revised the sentence “[EMIM]+ as a cationic surfactant, its hydrophobicity could reduce the surface tension of solvent, regulating the formation of nanoparticles.” to “the hydrophobicity of [EMIM]+ rooted in alkyl side chains …...” Please see page 3 paragraph 4 line 5-7 (the automatically generated line 137-139).
- About the problem for that “[BMIM]+ cation could interact with diethanolamine anion, which slowed down the diffusion rate of nickel cation and the decomposition rate of red phosphorus, increasing the porosity and pore size of as-prepared catalyst.” is incorrect. Diethanolamine cannot be an anion.
We are very sorry for this error. We have revised the sentence “[BMIM]+ cation could interact with diethanolamine anion……” to “[BMIM]+ cation could interact with phosphorous anion induced by diethanolamine……”. Please see page 4 paragraph 2 line 4-5 (the automatically generated line 164-165).
- About the problem for that “Recently, our research group have explored the synthesis method of nickel phosphides based on quarter phosphonium salt [40]” instead of “quarter” should be “quaternary”.
Thank you very much. We have revised “quarter” to “quaternary”. Please see page 6 paragraph 3 line 2 (the automatically generated line 250).

Reviewer 2 Report
The authors reviewed the application of ionic liquid/deep eutectic solvent-mediated Ni-based catalysts for water splitting. The submission can be accepted after revision considering the following points:-
1. The title should be revised to be clear, precise, and informative. ‘Electrocalysis’ should be added to the title.
2. ‘Considering the rapid developments in this topic, the achievements of IL- and DES-mediated Ni-based catalysts presented here are until June 2022.’ Please, update the literature and date.
3. Literature should be summarized in Tables.
4. In-depth discussion should be involved.
5. The authors should obtain permission for Figures reprinted from literature.
6. References for ionic liquids in the Introduction should be updated, including these References; 10.1039/c4tb00528g; https://doi.org/10.1016/j.molliq.2022.118556; 10.1016/j.aca.2012.12.054
7. The language should be revised, and typos should be corrected.
Author Response
Aug. 12, 2022
Dr. Prof. Reviewer 2
Thank you very much for your letter on Thu., Aug 4, 2022, which were about our manuscript entitled “Ionic liquid/deep eutectic solvent-mediated Ni-based catalysts and their application in water splitting electrocatalysis” (catalysts-1856033). We are very grateful to you for the valuable comments and suggestions. We have revised the manuscript. Any revisions to the manuscript are highlighted in yellow color, such that any changes can be easily viewed by you. A list of changes and answers to the comments are given in the following pages. In the list, we marked the page and line number to facilitate reviewers to review. However, it should be noted here that the automatically generated line number in the template is different on different computers. What we have marked may be different from the corresponding ones on your computer. We are very sorry for this.On behalf of the coauthors, I hope that you are satisfied with the changes and look forward to a possible further publication process. In case of any question, please do not hesitate to contact me. Thank you very much.
Sincerely yours,
Prof. Dr. Bingwei Xin
xinbingwei@dzu.edu.cn
Dezhou University
Response to the comments and changes to the manuscript
Comments and Suggestions for Authors
The authors reviewed the application of ionic liquid/deep eutectic solvent-mediated Ni-based catalysts for water splitting. The submission can be accepted after revision considering the following points:-
- The title should be revised to be clear, precise, and informative. ‘Electrocalysis’ should be added to the title.
Answer:
Thank you very much. In the revised revision, the title has been revised to “Ionic liquid/deep eutectic solvent-mediated Ni-based catalysts and their application in water splitting electrocatalysis”, in order to be more precise, and more informative.
- ‘Considering the rapid developments in this topic, the achievements of IL- and DES-mediated Ni-based catalysts presented here are until June 2022.’ Please, update the literature and date.
Answer:
In the revised manuscript, we have updated the literatures to what are now available, about until August. So we have revised the sentence “……the achievements of IL- and DES-mediated Ni-based catalysts presented here are until June 2022.” to “……presented here are until August 2022”. Please see page 3 paragraph 1 line 1 from the bottom (The automatically generated line 111).
To reflect the latest achievements in this field, the updated literatures have been referenced, described and discussed in revised manuscript, which are numbered as Ref. 4, 10, 15, 55, 70, respectively. For example, we have detailed descriptions of Ref. 55 and 70. Please see last paragraph of page 10 and page 11, respectively.
Finally, the newly added literatures have been listed in the “References” section highlighted in yellow color.
- Literature should be summarized in Tables.
Answer:
Thank you very much. We have summarized the literatures in Table 1 and Table 2. Please see page 7 and 15, respectively.
- In-depth discussion should be involved.
Answer:
Thank you very much for your comments. On the one hand, we further discuss the research status and development trends in the revised manuscript, providing a relative in-depth insight, new insights, understanding. The corresponding modifications are highlighted in yellow color in the revised manuscript. On the other hand, we added a table in “Summary and outlook” section to further in-depth discussion of the advantages of IL and/or DESs over traditional methods. Please see Table 3 in Page 17.
- The authors should obtain permission for Figures reprinted from literature.
Answer:
Thank you very much. We have obtained permission for Figures reprinted from literatures, and have sent them to editor with email attachment.
- References for ionic liquids in the Introduction should be updated, including these References; 1039/c4tb00528g; https://doi.org/10.1016/j.molliq.2022.118556; 10.1016/j.aca.2012.12.054.
Answer:
Thank you very much. We have updated these References in the “Introduction” section with description. 10.1039/c4tb00528g, https://doi.org/10.1016/j.molliq.2022.118556, and 10.1016/j.aca.2012.12.054 have been numbered as 14, 15, 16, respectively. Please see the page 2 paragraph 2 line 6 (The automatically generated line 61).
- The language should be revised, and typos should be corrected.
Answer:
Thank you very much. We have made carefully English grammar revisions in the revised manuscript, and have corrected the typos.

Reviewer 3 Report
Zhang et al reported the Ionic liquid/deep eutectic solvent-mediated Ni-based catalysts 2 and their application in water splitting. The review is well drafted. However, some major issues need to solve before its publication as below.
1. Introduction has a number of small paragraphs. While reading its disturbing. It is suggested to combine the small paragraphs.
2. Why is authors only concentrated on Ni based catalysts alone? does it mean Ionic liquid/deep eutectic solvent synthesis is not adopted for other transition metal catalysts?
3. Please point out specifically why Ionic liquid/deep eutectic solvent are better than the traditional synthesis processes.
4. List our what type of Ionic liquid/deep eutectic solvent have been used and their catalyst morphology, synthesis conditions, OER and HER kinetics etc in a tabular column.
5. Is there any studies that showed the comparison of Ionic liquid/deep eutectic solvent and traditionally synthesis process, together? Can you spot the differences?
6. Based on the review could you make a conclusive paragraph on which type of Ionic liquid/deep eutectic solvent are the best and why. This help researchers to choose the best one.
7. Authors needs to find out the gaps and future research direction in this field.
8. Some comparative table (advantages vs disadvantages) is needed to understand the advantages of Ionic liquid/deep eutectic solvent synthesis over rational methods such as precipitation method/sol-gel method/solvothermal/hydrothermal method etc. This helps the young researchers to gain basic knowledge on the Ionic liquid/deep eutectic solvent method.
9. Improve the figure quality. Additional attractive/meaningful images can be included.
Author Response
Aug. 12, 2022
Dr. Prof. Reviewer 3
Thank you very much for your letter on Thu., Aug 4, 2022, which were about our manuscript entitled “Ionic liquid/deep eutectic solvent-mediated Ni-based catalysts and their application in water splitting electrocatalysis” (catalysts-1856033). We are very grateful to you for the valuable comments and suggestions. We have revised the manuscript. Any revisions to the manuscript are highlighted in yellow color, such that any changes can be easily viewed by you. A list of changes and answers to the comments are given in the following pages. In the list, we marked the page and line number to facilitate reviewers to review. However, it should be noted here that the automatically generated line number in the template is different on different computers. What we have marked may be different from the corresponding ones on your computer. We are very sorry for this.On behalf of the coauthors, I hope that you are satisfied with the changes and look forward to a possible further publication process. In case of any question, please do not hesitate to contact me. Thank you very much.
Sincerely yours,
Prof. Dr. Bingwei Xin
xinbingwei@dzu.edu.cn
Dezhou University
Response to the comments and changes to the manuscript
Zhang et al reported the Ionic liquid/deep eutectic solvent-mediated Ni-based catalysts and their application in water splitting. The review is well drafted. However, some major issues need to solve before its publication as below.
- Introduction has a number of small paragraphs. While reading its disturbing. It is suggested to combine the small paragraphs.
Reply:
I'm very sorry to make the reviewers feel disturbing due to a number of small paragraphs. In this revised version, we have reduced the “Introduction” section into five paragraphs. The first paragraph analyzes water splitting and its electrocatalysts. The second paragraph describes ionic liquids (ILs). The third paragraph discusses the deep eutectic solvents (DESs). The fourth paragraph offers the advantages of ILs and DESs in catalyst preparation. The fifth paragraph provides an overview of the content of this review.
- Why is authors only concentrated on Ni based catalysts alone? does it mean Ionic liquid/deep eutectic solvent synthesis is not adopted for other transition metal catalysts?
Reply:
Our research work has been mainly focused on the research of IL/DES-derived nickel-based catalysts, so we are relatively familiar with this aspect. This does not mean that the synthesis of other transition metal catalysts in IL and DES is not advantageous. Like nickel-based catalysts, their synthesis research has been carried out very much, and many advanced results have been achieved.
- Please point out specifically why Ionic liquid/deep eutectic solvent are better than the traditional synthesis processes.
Reply:
The special liquid structure and special properties of ILs and DESs determine their advantages over traditional synthetic routes in the catalyst synthesis process. Their advantages are as follows:
ILs and DESs are almost nonvolatile and have highly thermal stability. Therefore, they can maintain normal pressure or even vacuum at high temperature in reactions, which not only reduce the production of toxic gases but also avoid the danger of high-pressure reactions; ILs and DESs can dissolve materials that are difficult to dissolve in conventional solvents, such as metals, metallic oxides and so on, so that the reactions can proceed homogeneously; ILs and DESs have special physicochemical properties and structures. On the one hand, ILs and DESs have greater viscosity, higher polarity, lower surface tension than aqueous solution or conventional organic solvents. On the other hand, they can spontaneously form extended hydrogen bonds. Therefore, ILs and DESs can regulate the morphology of inorganic nanomaterials, while tune the nucleation and growth of catalysts; the microwave absorption and large electrochemical window of ILs and DESs make them suitable for microwave and electrodeposition synthesis. Various nickel-based catalysts have been successfully prepared in ILs and DESs. Please see page 3 paragraph 4 line 2-14 (The automatically generated line 120-132) and page 8 paragraph 1 line 1-2 (The automatically generated line 295-296).
- List our what type of Ionic liquid/deep eutectic solvent have been used and their catalyst morphology, synthesis conditions, OER and HER kinetics etc in a tabular column.
Reply:
Thank you very much for the valuable suggestions.
We have summarized the literatures in Table 1 and Table 2, including their catalyst morphology, synthesis conditions, OER and HER kinetics (Tafel slopes). Please see page 7 and 15, respectively.
- Is there any studies that showed the comparison of Ionic liquid/deep eutectic solvent and traditionally synthesis process, together? Can you spot the differences?
Reply:
Numerous studies have shown that ionic liquids/deep eutectic solvents are superior to traditional synthetic processes in the preparation of transition metal catalysts. In this review, we also make extensive comparisons. We spot the differences with two typical examples.
Typical example 1: In 2018, Roberts compared the differences between ionic liquids and oleylamine as solvents under the same conditions. It is found that oleylamine acts as solvent to offer the mixture of Ni2P and Ni12P5. However, [BMIM]Tf2N as a medium induced the preparation of pure Ni2P nanocrystals. This is because different media lead to different growth processes of nickel phosphides. In 1-ethyl-3-methylimidazolium bis(trifluoromethylsulfonyl)imide ([BMIM]Tf2N), [BMIM]+ cations were adsorbed on the surface of Ni2P crystal nuclei, which in turn affected the nucleation and growth of phosphides. Nickel phosphides had no crystal phase transformation during the growth process so that a single crystal phase was obtained. However, non-polar octadecene couldn't be adsorbed on the surface of nickel phosphides, undergoing a transformation from Ni12P5 to Ni2P during the phosphating process. Therefore, IL could effectively adjust the crystal phase of nickel phosphides and optimized the catalytic performance. Please see page 4 the last paragraph, referenced Ref. 40.
Typical example 2: The researchers compared the morphology and catalytic performance of Ni nanoparticles prepared by electrodeposition in acetate buffer and ChCl/EG DES electrolytes, respectively. By comparison, it was found that the electrodeposition process in DES was more efficient than that of acetate buffer, producing Ni nanoparticles with more uniform needle-shaped structure and smaller size. Therefore, DES-mediated Ni nanoparticles exhibited better electrocatalytic performance, which required overpotentials both HER and OER of 154 mV and 350 mV to achieve 10 mA cm−2, respectively with smaller Tafel slopes. Whereas, Ni from acetate required overpotentials of 164 mV and 400 mV for HER and OER. Please see page 10 the last paragraph, referenced Ref. 55.
- Based on the review could you make a conclusive paragraph on which type of Ionic liquid/deep eutectic solvent are the best and why. This help researchers to choose the best one.
Reply:
Different types of ILs and DESs have different properties, such as viscosity, density and conductivity, and so on. The commonly used ILs are mainly imidazolium-type ILs. The preparation, purification methods, properties, and liquid structures of such kind of ILs have been intensively studied in both theoretical and practical area. However, owing to increasing doubts about the environmental benign of ILs, DESss have a tendency to replace ILs. Common DESs are choline chloride (ChCl)-based DESs. With the application of IL and DES in the field of catalyst preparation, using their designability to design IL and DES with novel composition and superior performance will be the development direction. The “all-in-one” strategy will promote ILs and DESs to participate in the reaction as carbon source, nitrogen source and so on as well media and templates. In order to achieve this goal, researchers have been working on this research. Wegner et al. have reviewed in detail the interdependencies of ILs with the metal nanoparticle and tried to elucidate influences of the IL-cation, -anion and alkyl chain length [(10.1007/s41061-017-0148-1)].
- Authors needs to find out the gaps and future research direction in this field.
Reply:
ILs and DESs have been involved in the preparation of transition metal catalysts for many years, and great achievements have been made in academics. However, there are still some urgent problems to be solved. First, the preparation of catalysts using ILs and DESs requires further theoretical studies, including the nucleation and growth processes of metal catalysts induced by them, which will be the theoretical basis to realize controllable preparation of catalysts. Second, “all-in-one” synthesis strategy is needed to further studied. The biggest challenge of this strategy is how to creatively select ILs and DESs to get target products with excellent performance. This method brings practical benefits to the improvement of technology and cost saving, helping people achieve carbon peaking and carbon neutralization. In addition, few achievements have actually been applied for industrialization now.
In the “Summary and outlook” section, we have further explored the gaps and future research directions. Please see page 18, two paragraphs below the Table 3.
- Some comparative table (advantages vs disadvantages) is needed to understand the advantages of Ionic liquid/deep eutectic solvent synthesis over rational methods such as precipitation method/sol-gel method/solvothermal/hydrothermal method etc. This helps the young researchers to gain basic knowledge on the Ionic liquid/deep eutectic solvent method.
Reply:
Thank you very much. We have listed a comparative table to compare advantages and disadvantages between ILs/DESs and traditional synthesis. Please see Table 3 in Page 17.
- Improve the figure quality. Additional attractive/meaningful images can be included.
Reply:
Thank you very much. We have redrawn Figure 1 (page 4), Figure 3 (page 7) and Figure 4 (page 10) to improve the quality.

Round 2
Reviewer 3 Report
The authors have substantially improved the paper more than I expected. Therefore, the reviewer thinks its suitability for publication in this journal, in the present form.